# The Complete Mitochondrial Genome of the Chinese White Wax Scale Insect, *Ericerus pela* Chavannes (Hemiptera: Coccidae), with Novel Gene Arrangement and Truncated tRNA Genes

**DOI:** 10.3390/insects14030290

**Published:** 2023-03-16

**Authors:** Jia-Qi An, Shu-Hui Yu, Shu-Jun Wei, Hong-Ping Zhang, Yuan-Chong Shi, Qiu-Yu Zhao, Zuo-Yi Fu, Pu Yang

**Affiliations:** 1Institute of Highland Forest Science, Chinese Academy of Forestry, Kunming 650224, China; 2College of Biology and Environment, Nanjing Forestry University, Nanjing 210037, China; 3College of Agriculture and Life Sciences, Kunming University, Kunming 650214, China; 4Institute of Plant Protection, Beijing Academy of Agriculture and Forestry Sciences, Beijing 100097, China; 5Key Laboratory of Breeding and Utilization of Resource Insects of National Forestry and Grassland Administration, Kunming 650224, China

**Keywords:** Coccoidea superfamily, mitochondrial genome, PacBio sequencing, gene rearrangement

## Abstract

**Simple Summary:**

*Ericerus pela* Chavannes (Hemiptera: Coccoidea) is one of the most economically valuable resource insects in China. Its mitochondrial genome provides essential information for the molecular identification and genetic study of this species. In this paper, we assembled the mitochondrial genome of *E. pela*. Gene rearrangement analysis showed that compared to other scale insects, *E. pela*’s *atp6*, *atp8*, and a few tRNAs were obviously rearranged. Moreover, there were nine tRNAs identified to have obvious truncated structures. Synteny analysis based on the whole mitochondrial genomes showed that a significant rearrangement of homologous blocks had occurred within this Coccoidea species. The results revealed great variations in the intergenic regions among Coccoidea species and novel gene arrangement and truncated tRNA genes in the mitochondrial genome of *E. pela*.

**Abstract:**

The Chinese white wax scale insect, *Ericerus pela* Chavannes (Hemiptera: Coccidae), is one of the scale insects with great economic value and has been dispersed and reared in China for over one thousand years. Its mitochondrial genome provides essential information for the molecular identification and genetic study of this species. We assembled the complete mitochondrial genome of *E. pela* based on PacBio sequencing and analyzed its genomic features. The genome was 17,766 bp in length with 13 protein-coding genes, 22 tRNAs, and two rRNA genes. The analysis results showed *E. pela* had significant gene rearrangements involving tRNAs compared with other Coccoidea species. Furthermore, *E. pela*’s nine tRNAs were identified to have obvious truncated structures. The phylogenetic tree compiled of the species showed a long branch of the Coccoidea lineage, which indicated the high evolutionary rate in this group. Our study revealed the mitochondrial characteristics of *E. pela* and enriched the mitochondrial genetic information on Coccoidea species. It also determined the occurrence of gene rearrangement for the species in this superfamily.

## 1. Introduction

The scale insects from the superfamily Coccoidea of Hemiptera feed on plant sap with their piercing–sucking mouthparts [1,2]. As reflected in their common name, scale insects always possess a protective cover or a cuticle “scale” such as a waxy layer secreted during the insect’s lifecycle [3,4,5]. The superfamily Coccoidea comprises 36 families containing approximately 8000 species [6,7,8,9]. All of these species can be informally divided into the monophyletic neococcoids and the uncertain archaeococcoids [10].

Compared to the nuclear genome, the mitochondrial genome has been seen as an ideal research system due to its smaller genome size, comparably stable gene composition, and higher evolutionary rate [11]. Moreover, mitochondrial genomics has been applied in fields of study including biogeography, molecular evolution, hybridization, and especially, phylogenetics [11,12]. However, for the group of hemipteran insects containing more than 83,000 species, just hundreds of complete mitochondrial genomes have been reported [13]. As of May 2022, only ten nuclear genomes (eight were preliminary assemblies without annotations) and nine complete mitochondrial genomes (only three have complete gene annotations) of Coccoidea have been released on the NCBI [1,2,14]. Furthermore, for the Coccidae group (soft-scale insects), the complete mitochondrial genomes published on PubMed exist only for *Ceroplastes japonicus* [15] and *Saissetia coffeae* [16]. Clearly, more molecular information on scale insects is needed to understand the evolution of their mitochondrial genomes.

Most Coccoidea species are agriculture and forestry pests, except for very few species with a high economic value, such as the Chinese white wax scale insect *Ericerus pela* (Chavannes, 1848) [17]. Previous work has related to *E. pela*’s use in biological research, molecular research, and the popularization of relevant applications [18,19,20,21,22,23,24,25,26]. Regarding its breeding history, people initially started to collect white wax as early as approximately the 3rd century AD in China [26]. The wax produced by insects is an important commodity that has been widely applied in cosmetics, Chinese traditional medicine, the manufacturing industry (e.g., automotive and furniture manufacture), and the printing industry for more than one thousand years [27,28]. In addition, compared to other wax products, Chinese white wax has the highest content of long-chain alcohols (hexacosanol, tetracosanol, triacontanol, and policosanol), which have been widely applied in the pharmaceutical industry. For instance, tetracosanol is a raw pharmaceutical material used for treating hypercholesteremia, and policosanol is often used to lower cholesterol and triglyceride levels [29].

However, we still lacked molecular information on *E. pela* for the further detection of its phylogenetic affiliation. There has been an increasing trend in observing hemipteran species’ mitochondrial gene rearrangements, which has involved species belonging to Aleyrodoidea, Coccoidea, and Fulgoroidea [30,31]. Moreover, for the only two published mitochondrial genomes of the Coccidae family, both *C. japonicus* [15] and *S. coffeae* [16] have identified tRNAs with truncated secondary structures. Whether *E. pela*’s mitochondrial genome had these specific features still remained uncertain. Considering these unsolved scientific questions, it was suitable for us to comprehensively investigate *E. pela*’s mitochondrial genome to find out its genetic variations to not only other scale insect species but also outgroup insect species.

In this study, we assembled the complete mitochondrial genome of *E. pela* and conducted a phylogenetic analysis for related hemipteran insects. This study has great significance for determining *E. pela*’s phylogenetic position and revealing comprehensive genetic differences between scale insects and other insects.

## 2. Methods

### 2.1. Sample Preparation and DNA Extraction

We used male adults of *E. pela* reared at the Institute of Highland Forest Science in Kunming City (25°05′ N, 102°72′ E), Yunnan Province, for the mitochondrial genome sequencing. About 30 male adults were collected for DNA extraction.

The samples were first processed with double distilled water and liquid nitrogen. After preliminary lysis with lysis buffer (Tris-HCl, [EDTA]-2Na, guanidine hydrochloride, and NaCl), proteinase K, and sodium dodecyl sulfate (SDS), the sample was incubated at 56 °C (for 45 min). Then, genomic DNA was extracted by isopropyl alcohol (Tris-phenol, chloroform, and isoamyl alcohol) precipitation by centrifugation at 4700 rpm (10 min). After transferring the supernatant into a new tube, the isolation step was repeated another time. Isopropyl alcohol was added to obtain a DNA precipitate. A concentration of 70% (*v*/*v*) ethanol was added to wash the obtained DNA precipitate. After removing the ethanol, the DNA precipitate was dissolved with Tris-EDTA. RNase A was used for RNA degradation. DNA concentration was determined using a Qubit fluorometer (Invitrogen, Carlsbad, CA, USA). Pulsed-field gel electrophoresis was used to detect DNA quality [19]. The samples were highly qualified for the next library construction.

### 2.2. Library Construction and Sequencing

The genomic DNA of *E. pela* was disrupted and fragments of 13–16 Kb were selected. The two ends of each fragment were ligated with a single string circle to form an SMRTbell. The PacBio HiFi library was constructed.

The library was sequenced on the PacBio platform. The obtained data were processed for the next genome assembly.

### 2.3. Genome Assembly

The PacBio sequencing data were trimmed as follows: (1) Remove adapter contamination; (2) remove low-quality reads; and (3) remove reads with a certain proportion of Ns (5% as default) [32]. Subreads were filtered to obtain circular consensus sequence reads. To assemble the mitochondrial genome of *E. pela*, four conserved mitochondrial gene fragments, which included three cytochrome c oxidase genes (*cox1, cox2, cox3*) and one cytochrome b reductase gene (*cob*), were used to filter the PacBio circular consensus sequence reads. Then, the filtered mitochondrial circular consensus sequence reads were used for mitochondrial genome assembly with Racon and Miniasm [33,34,35].

### 2.4. Mitochondrial Genome Annotation

Protein-coding genes (PCGs), rRNA, and tRNA genes were annotated by using the MFannot webserver (https://megasun.bch.umontreal.ca/apps/mfannot/, accessed on 20 April 2022) and the MITOS webserver (with the advanced setting of the protein prediction method used by Al Arab et al. [36]) (http://mitos2.bioinf.uni-leipzig.de/index.py, accessed on 20 April 2022). Intergenic spacer sequences and gene overlap were obtained through manual curation. The complete mitochondrial genome circular map was made by using the webserver Proksee (https://proksee.ca/, accessed on 20 April 2022).

### 2.5. Gene Component and Structural Analysis

The secondary structures of the 13 tRNA genes were predicted with the MITOS webserver. The remaining nine tRNA genes could be roughly predicted by the sequence alignment depicted by the RNA Folding Form V2.3 of the UNAFold webserver (http://www.unafold.org/mfold/applications/rna-folding-form-v2.php, accessed on 20 April 2022), since these tRNAs could not be depicted by either the MITOS webserver or the tRNAscan-SE search server. The relative synonymous codon usage (RSCU) of the protein-coding genes of *E. pela* was analyzed by using the software CodonW (v1.4.2) (https://codonw.sourceforge.net/, accessed on 20 April 2022). The base composition skew was analyzed by manual calculation. The calculations were performed according to the formula GC skew = [G − C]/[G + C] and the formula AT skew = [A − T]/[A + T] [37].

### 2.6. Phylogenetic Analysis

The phylogenetic tree was constructed based on the 13 mitochondrial PCGs’ amino acid sequences of 15 hemipteran species (Table 1). Among the 15 hemipteran species, fourteen species were all from Homoptera, which comprised four species chosen from within the superfamily Coccoidea, three species chosen from within the superfamily Psylloidea, three species chosen from within the superfamily Aphidoidea, three species chosen from within the superfamily Aleyrodoidea, and one species selected from the superfamily Cicadoidea. Moreover, except for the Cicadoidea species *Magicicada tredecassini*, which belongs to the suborder Auchenorrhyncha, the other 13 species were all from the suborder Sternorrhyncha. The heteropteran species *Lethocerus deyrollei* was selected as the outgroup. All the complete mitochondrial genomes of the species mentioned above were downloaded from the NCBI.

Amino acid sequence alignment was performed in MUSCLE [38] (https://www.ebi.ac.uk/Tools/msa/muscle/, accessed on 20 April 2022) with the default parameters. The alignment results were manually converted to the FASTA format feasible for the construction of the phylogenetic tree and then imported into the IQ-TREE webserver [39] (http://iqtree.cibiv.univie.ac.at/, accessed on 20 April 2022). The model was calculated by the application “model selection” in IQ-TREE and the one with the lowest score of the Bayesian information criterion (BIC) was adopted in the study. The model applied in the interspecies phylogenetic analysis was “mtZOA+F+I+G4”. The phylogenetic test was performed using the bootstrap method after setting the number of bootstrap replications to be 1000. The phylogenetic tree was constructed using the maximum likelihood (ML) statistical method.

### 2.7. Gene Rearrangement Analysis

The annotated mitochondrial genomes of *Didesmococcus koreanus* (Hemiptera: Coccidae), *Aclerda takahashii* (Hemiptera: Aclerdidae), *Saissetia coffeae* (Hemiptera: Coccidae), *E. pela* (Hemiptera: Coccidae), *Aphis glycines* (Hemiptera: Aphididae), and *Drosophila melanogaster* (Diptera: Drosophilidae) were selected for gene rearrangement analysis. The GenBank files of the selected species that contained information on the mitochondrial gene orders were downloaded from the NCBI (https://www.ncbi.nlm.nih.gov/, accessed on 20 April 2022); the situation of the rearrangements that had taken place within the species was organized according to the GenBank files.

### 2.8. Synteny Analysis

Synteny analysis was carried out with *E. pela* and the nine Coccoidea species whose genomes had been released by the NCBI (Table 2). The mitochondrial genome data of the nine Coccoidea species were downloaded from the NCBI. After importing the sequences into the software in FASTA format, the analysis was performed at the nucleic acid level with Mauve (v2.1.1) (https://darlinglab.org/mauve/mauve.html, accessed on 20 April 2022) with the default parameters.

## 3. Results

### 3.1. General Genomic Features

The complete mitochondrial genome of *E. pela* assembled with PacBio reads was 17,766 bp in length (GenBank accession number: ON055488). The base composition was 38.78% A, 49.43% T, 4.00% C, and 7.79% G. It had an overall G+C content of 11.79%. From the annotation result, this complete mitochondrial genome contained 13 protein-coding genes (PCGs), 22 transfer RNA genes (tRNAs), two ribosomal RNA genes (rRNAs), and one control region (CR) (Figure 1).

The 13 annotated PCGs included the three cytochrome c oxidase (*cox*) genes *cox1*, *cox2*, and *cox3,* one cytochrome b reductase gene *cob,* two ATP synthase synthesis genes *atp6* and *atp8,* and seven NADH dehydrogenase genes *nad1*, *nad2*, *nad3*, *nad4*, *nad4L*, *nad5*, and *nad6*.

In addition to the 13 PCGs, there were 24 essential genes without coding protein in the mitochondrial genome, which included 22 transfer RNA genes and two ribosomal RNA genes (*rrnL* and *rrnS*). Specifically, only 13 tRNA genes were identified and depicted by either the MITOS webserver or the tRNAscan-SE search server (Table 3).

There were six overlaps between 12 adjacent genes varying from 1 bp to 41 bp; the overlaps were 86 bp in total. Only one control region (CR) was located between *trnM* and *atp8*, which contained a possible intergenic spacer region.

### 3.2. Gene Rearrangement

According to the gene rearrangement analysis results, three (*D. koreanus*, *S. coffeae*, and *A. takahashii*) of the four scale insects had a general conserved gene order, except for some tRNAs between *rrnS*-*nad6* and *nad5-nad3*. For the four Coccoidea species, we marked the three most conserved gene regions with black lines (*nad1*-*nad2*, *nad6*-*nad5*, and *nad3*-*cox1*) and the three most divergent regions of *E. pela* with red arrows (Figure 2). As the representative hemipteran species and the model species of insect, the gene orders of *A. glycines* and *D. melanogaster* were used to make a comparison. In general, the gene rearrangement of *E. pela* occurred in three regions: for the obvious insertions between *rrnS*-*nad1*; for the deletion between *nad5*-*nad3*; and between *cox1*-*trnL1.* Moreover, though *E. pela* had a generally conserved gene order of PCGs, *atp8*, which was dramatically moved to a position between *trnY* and *trnL1*, became the most significant difference among the six listed species, while in the other three Coccoidea species and even in *A. glycines* and *D*. *melanogaster, atp8* and *atp6* were adjacent genes. Furthermore, it was clear that the three generally conserved regions of Coccoidea insects were not available for *A. glycines* and *D. melanogaster*, which meant it was comparatively far away in phylogenetic distance between Coccoidea and Diptera. Additionally, even in the Hemiptera order, there already existed significant divergent within species.

### 3.3. Protein-Coding Genes and Codon Usage

The total length of the 13 PCGs was 10,686 bp, accounting for 60.15% of the whole genome. The A + T content of all PCGs was 87.92%. The A + T content and sequence length of each PCG varied distinctly, but the 13 PCGs all had an A + T content higher than 80%. The lowest A + T content value was found for the *cox1* gene (81.57%) and the highest value was found for the *nad6* gene (93.29%). Among these 13 PCGs, the *atp8* gene was the shortest, at 192 bp in length, while the *nad5* gene was the longest, at 1632 bp.

All the exons in the 13 PCGs of *E. pela* (except for stop codons) encoded a total of 3806 amino acids. There were two types of codons initiating transcription. The main one was ATN codons for all the PCGs (12 PCGs in total) except for *atp8* and the other one was AGA, which initiated the *atp8* gene.

Regarding the overall statistics of relative synonymous codon usage (RSCU), tyrosine (Tyr), isoleucine (Ile) and phenylalanine (Phe) could be the most common amino acids in the whole mitochondrial genome. Ile accounted for the largest proportion (17.87%), followed by Tyr (16.76%) and Phe (12.74%), while glutamine (Gln) accounted for the smallest proportion (0.3%). Representing all the codons, UAU of Tyr, AUU of Ile, and UUU of Phe were the codons with the highest frequencies (Figure 3).

### 3.4. rRNAs and tRNAs

The mitochondrial genome of *E. pela* had two adjacent ribosomal RNA genes, which included a small subunit ribosomal RNA gene (*rrnS*) and a large subunit ribosomal RNA gene (*rrnL*). The lengths of the two ribosomal genes were 599 bp and 733 bp, with A + T content for rrnS of 85.81% and for *rrnL* of 87.03%.

There were 22 tRNA genes in the whole mitochondrial genome, which comprised 13 tRNA genes identified by MITOS (http://mitos2.bioinf.uni-leipzig.de/index.py, accessed on 20 April 2022) and nine truncated tRNA genes recognized by aligning homologous sequences with *E. pela*’s mitochondrial genome through DNAMAN (v7.0) (https://www.lynnon.com/dnaman.html, accessed on 20 April 2022). In total, 20 standard amino acids were conveyed by all the 22 tRNAs. For the 13 identified genes, only four tRNAs (*trnK*, *trnL2*, *trnM*, and *trnI*) exhibited a typical cloverleaf secondary structure, which contained four arms and one loop. In addition to these four ordinary tRNAs, the remaining tRNAs all had irregular secondary structures that lacked specific arms (Figure 4).

### 3.5. Phylogenetic Analysis

In the phylogenetic analysis, the 13 species were clustered into four groups that exactly represented the four hemipteran superfamilies of the suborder Sternorrhyncha (Coccoidea, Aphidoidea, Aleyrodoidea, and Psylloidea). The Aphidoidea superfamily formed a sister group with the Coccoidea superfamily, which meant that the two superfamilies had a close genetic relationship. Compared with the other three hemipteran superfamilies, it was clear that the superfamily Psylloidea has a closer genetic relationship with the suborder Auchenorrhyncha, which also meant that the suborder Sternorrhyncha has a relatively more distant relationship with the suborder Heteroptera than the suborder Auchenorrhyncha (Figure 5). Obviously, the suborder Heteroptera represented by *L. deyrollei* had the farthest phylogenetic distance with the Coccoidea superfamily.

### 3.6. Synteny Analysis

The synteny analysis showed that the total number of homologous blocks in these 10 Coccoidea species varied from 3 (*Drosicha corpulenta*) to 13 (*S. coffeae*) (Figure 6). The homologous blocks of each species exhibited significant distinctions in both size and relative position. *C. japonicus* and *C. rubens* exhibited an entire synteny relationship regarding their homologous blocks, while the other species all showed a certain extent of rearrangement of homologous blocks. In particular, *E. pela* had six homologous blocks in all, among which four homologous blocks were in common with *C. japonicus.* Additionally, *C. rubens* and *A*. *takahashii* both had four homologous blocks in common with *E. pela*.

## 4. Discussion

### 4.1. High A + T Bias

For insect mitochondrial genomes, it is common to have a significant A−T bias in the entire nucleotide’s composition. According to this phenomenon, scientists have proposed that since A–T base pairs have a lower nitrogen requirement than C–G base pairs, this significant A−T bias of the scale insect could reflect a possible adaptation to the lack of nitrogen during long evolutionary processes [16]. Because the plant sap absorbed by the insect could be nitrogen-deficient, after a long evolutionary process, the scale insect could have formed an A:T pair bias. The nine previously Coccoidea mitochondrial genomes from the NCBI contained A + T contents ranging from 82.54% (*D. koreanus* NC_057479) to 87.52% (*C. rubens* MT677923) (Table 2). *E. pela* had an extremely biased A + T content of 88.21%, higher than all the other nine Coccoidea species, which indicated an adaptation to a significantly high requirement for nitrogen. This was in accordance with the existence of symbiotic fungi, which could also serve as a nitrogen provider in vivo *E. pela* [18].

In our study, both *nad3* and *nad6* had extremely high A + T contents (90.06% and 93.29%, respectively) (Table 3) and were originally missed by BLAST-p (they were successfully predicted by the Al Arab et al. method afterwards, which adopted hidden Markov models (HMMs) exhibiting greater phylogenetic awareness). Our study provided another successful precedent for predicting genes with high A + T contents [16,40,41].

### 4.2. tRNA

Significantly, only 13 tRNAs were successfully predicted in the annotation procedure while the other nine tRNAs were obtained through the multiple alignments with other homologous sequences. Most significantly, most tRNAs contained truncated sequences that would have resulted in their irregular secondary structures, without a regular dihydrouridine/ribothimidine arm (T/D-arm) or acceptor stem. Furthermore, this was not a specific event only observed in *E. pela*. The first two reports about Coccoidea species’ mitochondria found similar tRNA truncation in both *C. japonicus* and *S. coffeae* [15,16]. In fact, this tRNA truncation phenomenon was first illustrated in other metazoan species, such as arachnids, nematodes, and mites [42]. Although these tRNAs are truncated, their function could still be supported and complemented by cytoplasmic tRNAs and some transcription factors [43,44,45]. For example, a report about nematode mitochondria showed that an evolved extra elongation factor EF-Tu [46] could assist with the interaction between truncated tRNAs and conventional EF-Tu. Moreover, some studies have reported that the truncated tRNAs in the mitochondrial genome could work normally once their irregular structure was completed by hybridization with specific probes. Additionally, tRNAs encoded by the nuclear genome can enter the mitochondria to assist with certain functions. Therefore, although they contain incomplete tRNA structures or even have deficient tRNA components, the mitochondria of these species could still function normally. However, to this day it cannot be ascertained whether this phenomenon is universal to all Coccidae species. Therefore, more evidence regarding the mitochondrial genome in this group is still in desperate need.

### 4.3. Gene Rearrangement and Synteny Analysis

According to the analysis results of gene rearrangement, there was generally conserved gene content among the Coccoidea species and beyond this range [47,48]. Obviously, gene arrangements took place in *E. pela*, in accordance with what researchers mentioned in a report on *S. coffeae* [16]. Moreover, it was mostly tRNA variations that were involved in the rearrangement, while the regions involving PCGs were in the comparatively conserved position (except for *E. pela*’s *atp8*). Significantly, compared to the other three Coccoidea species, *E. pela* exhibited maximum rearrangement in gene order, which indicated the phylogenetic relationship among four selected Coccoidea species from another point of view. In general, the analysis results could also show the great distinction in gene orders between hemipteran insects and species of other categories (represented by *D. melanogaster*).

The results of the synteny analysis seemed to be somewhat contradictory to the conservation (e.g., PCGs) obtained from existing gene rearrangement analyses. We speculated that this was because the synteny analysis proceeded on the basis of the whole mitochondrial genome, including both genes and intergenic regions, whereas when analyzing the rearrangement of gene orders, we considered only PCGs, rRNA genes, and tRNA genes, excluding intergenic regions. Therefore, it might be the great extent of variation between intergenic regions that led to the nonsynteny among homologous blocks.

### 4.4. Phylogenetic Analysis

Coccidae were thought to have evolved more recently with aphids than whiteflies or plant lice. Previous research proved the existence of a phylogenetic relationship between Coccidae and other hemipteran insects based on fragments of mitochondrial genes [4,17]. In this study, our phylogenetic analysis based on the mitochondrial genome provided robust evidence to further support the phylogenetic relationships among these hemipteran insects.

## 5. Conclusions

In this study, we assembled the complete mitochondrial genome of *E. pela* and revealed its mitochondrial characteristics. Our analyses revealed the presence of novel gene arrangement and truncated tRNA genes. Our results also showed a high evolutionary rate in Coccoidea and will provide essential information for the genetic study of these species.

## Figures and Tables

**Figure 1 insects-14-00290-f001:**
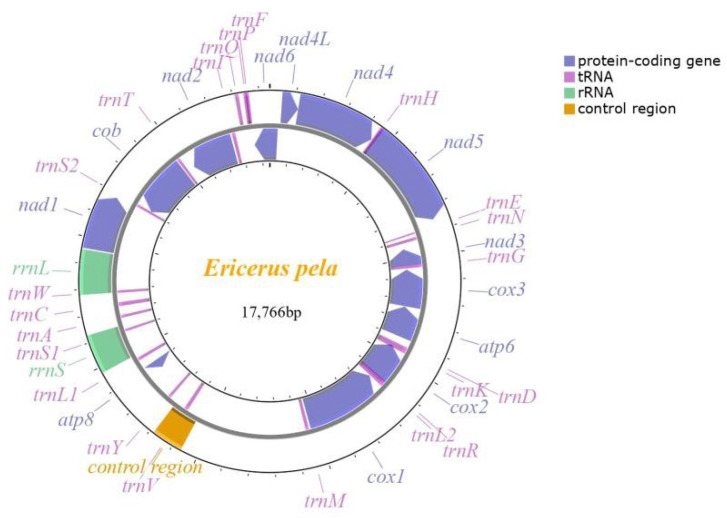
Mitochondrial gene map of *E. pela*.

**Figure 2 insects-14-00290-f002:**
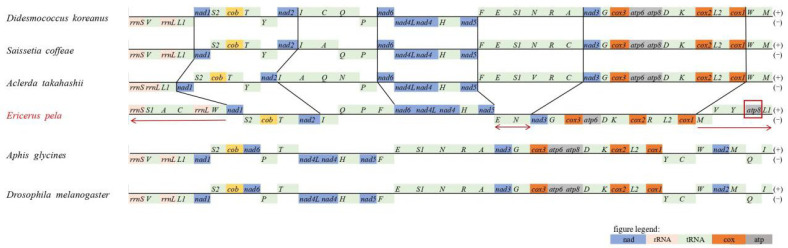
Mitochondrial gene order of 4 scale insects (Coccoidea)*, Aphis glycines* and *Drosophila melanogaster*. The genes on the H-strand (+) are shown above the line and the genes on the L-strand (−) are shown below the line. The black lines are used to show general conserved gene regions among the 4 Coccoidea species. The red arrows below *E. pela* represent the most significant distinctions.

**Figure 3 insects-14-00290-f003:**
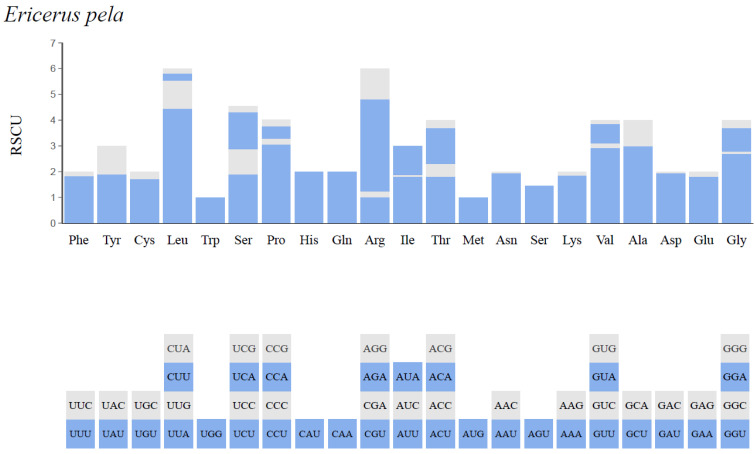
Relative synonymous codon usage (RSCU) of *E. pela*.

**Figure 4 insects-14-00290-f004:**
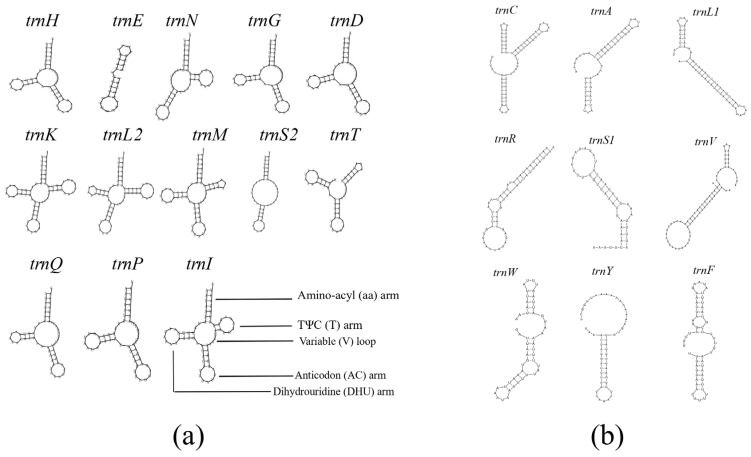
Secondary structures of *E. pela*’s tRNAs. The specific gene name of each tRNA is stated above each secondary structure. Dashes between bases represent hydrogen bonds. (**a**) Secondary structures depicted by MITOS and (**b**) secondary structures depicted by RNA Folding Form V2.3.

**Figure 5 insects-14-00290-f005:**
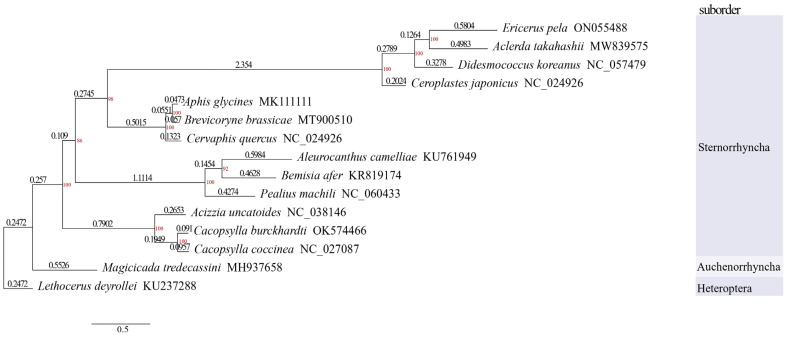
Phylogenetic tree based on amino acid alignments of 13 PCGs using the ML method. Bootstrap support values indicating the credibility of branches are shown on the nodes above the branches. The length of each branch represents the mitochondrial genetic variability of each species. Branch lengths are shown below the branches.

**Figure 6 insects-14-00290-f006:**
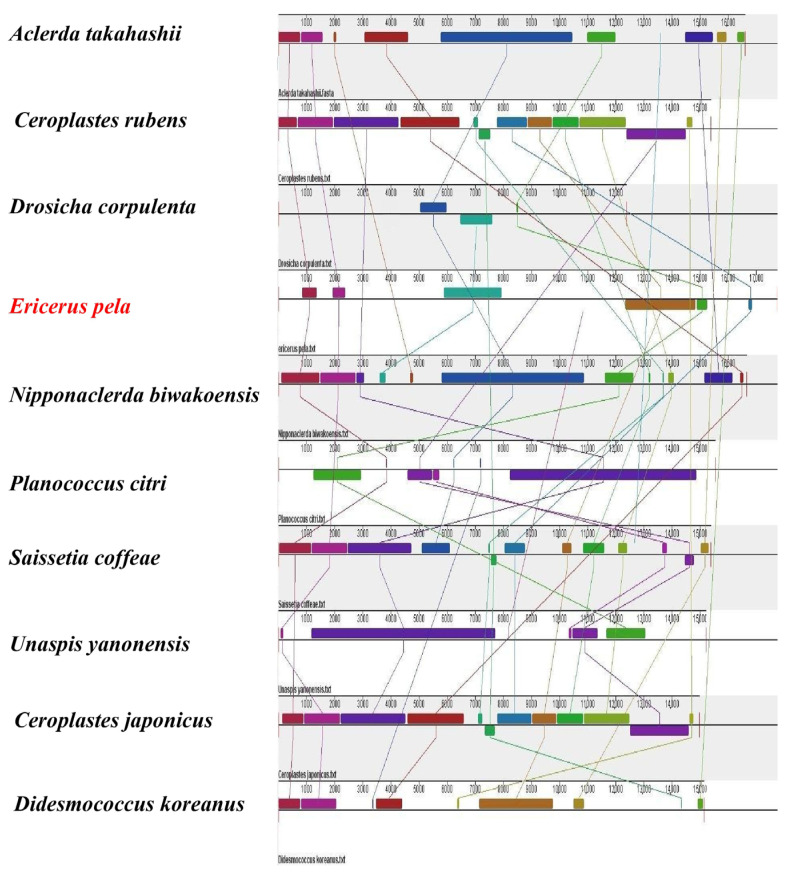
Synteny analysis based on whole mitochondrial genomes of 10 Coccoidea species. The solid-colored rectangles represent the homologous blocks among the selected species.

**Table 1 insects-14-00290-t001:** The species included in the phylogenetic analysis.

Species	Accession Number	Family	Suborder	Order
*Didesmococcus koreanus*	NC_057479	Coccidae	Sternorrhyncha	Hemiptera
*Aclerda takahashii*	MW839575	Aclerdidae	Sternorrhyncha	Hemiptera
*Ceroplastes japonicus*	MK847519	Coccidae	Sternorrhyncha	Hemiptera
*Ericerus pela*	ON055488	Coccidae	Sternorrhyncha	Hemiptera
*Cervaphis quercus*	NC_024926	Aphididae	Sternorrhyncha	Hemiptera
*Aphis glycines*	MK111111	Aphididae	Sternorrhyncha	Hemiptera
*Brevicoryne brassicae*	MT900510	Aphididae	Sternorrhyncha	Hemiptera
*Aleurocanthus camelliae*	KU761949	Aleyrodidae	Sternorrhyncha	Hemiptera
*Bemisia afer*	KR819174	Aleyrodidae	Sternorrhyncha	Hemiptera
*Pealius machili*	NC_060433	Aleyrodidae	Sternorrhyncha	Hemiptera
*Acizzia uncatoides*	NC_038146	Psyllidae	Sternorrhyncha	Hemiptera
*Cacopsylla burckhardti*	OK574466	Psyllidae	Sternorrhyncha	Hemiptera
*Cacopsylla coccinea*	NC_027087	Psyllidae	Sternorrhyncha	Hemiptera
*Magicicada tredecassini*	MH937658	Cicadidae	Auchenorrhyncha	Hemiptera
*Lethocerus deyrollei*	KU237288	Belostomatidae	Heteroptera	Hemiptera

**Table 2 insects-14-00290-t002:** The species included in the synteny analysis and their A + T%.

Species	Accession Number	A + T%
*Ceroplastes japonicus*	MK847519	85.15%
*Aclerda takahashii*	MW839575	84.51%
*Didesmococcus koreanus*	NC_057479	82.54%
*Saissetia coffeae*	MN863803	84.72%
*Unaspis yanonensis*	MT611525	86.57%
*Ceroplastes rubens*	MT677923	87.52%
*Drosicha corpulenta*	MK251061	87.21%
*Nipponaclerda biwakoensis*	MN193722	81.3%
*Planococcus citri*	MT611526	82.77%
*Ericerus pela*	ON055488	88.21%

**Table 3 insects-14-00290-t003:** Gene composition, Gene’s corresponding start/stop codons and AT skew.

Gene	Start	Stop	Length	Start/Stop Codons	A + T%
*nad4L*	191	460	270	ATT/TAA	88.52
*nad4*	462	1757	1296	ATA/TAA	89.74
*trnH*	1765	1820	56		
*nad5*	1786	3417	1632	ATA/TAA	89.58
*trnE*	3514	3543	30		
*trnN*	3605	3662	58		
*nad3*	3779	4120	342	ATT/TAG	90.06
*trnG*	4118	4176	59		
*cox3*	4201	4956	756	ATG/TAA	88.63
*atp6*	4964	5578	615	ATG/TAA	88.94
*trnD*	5716	5771	56		
*trnK*	5767	5835	69		
*cox2*	5795	6484	690	ATG/TAA	85.22
*trnR*	6459	6516	58		
*trnL2*	6503	6571	69		
*cox1*	6582	8117	1536	ATA/TAA	81.57
*trnM*	8151	8215	65		
Control region	8215	11163	2949		
*trnV*	10491	10563	73		
*trnY*	10885	10938	54		
*atp8*	11428	11619	192	AGA/AAA	85.94
*trnL1*	11786	11845	60		
*rrnS*	11911	12509	599		85.81
*trnS1*	12380	12425	46		
*trnA*	12642	12694	53		
*trnC*	12862	12936	75		
*rrnL*	13076	13808	733		87.03
*trnW*	13110	13160	51		
*nad1*	13830	14738	909	ATA/TAA	87.68
*trnS2*	14773	14822	50		
*cob*	14822	15901	1080	ATG/TAA	85.56
*trnT*	15922	15974	53		
*nad2*	16113	17018	906	ATT/TAA	90.73
*trnI*	17052	17116	65		
*trnQ*	17237	17295	59		
*trnP*	17367	17425	59		
*trnF*	17396	17450	55		
*nad6*	17451	146	462	ATA/TAA	93.29

## Data Availability

The genomic data in this study are available under the NCBI accession number: ON055488.

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
