# Peer review of "The Complete Mitochondrial Genome of the Chinese White Wax Scale Insect, Ericerus pela Chavannes (Hemiptera: Coccidae), with Novel Gene Arrangement and Truncated tRNA Genes"

_insects, 2023, doi:10.3390/insects14030290_

Round 1

Reviewer 1 Report

1. in the method part, Line 77, it is better to provide the location where the E. pela is collected.

2. in Figure 1, the length of the mitochondrial genome is 17,766bp, not 177,66bp.

3. line 185, the 3 most divergent regions

4. Line 188, for the obvious insertions between rrnS-nad1, and between cox1-trnL1, and the deletion between...

5. why do you label atp8 by redbox in Figure 2?

6. in the discussion part, since the nuclear genome of E. pela has been published, is there any similar finding that A+T content also increased in the nuclear genome? 

Author Response

Response to Reviewers

Dear Reviewers:

We appreciate your comments regarding our manuscript. Based on your suggestions, we amended the relevant parts of the manuscript as much as possible. Point-by-point responses to your comments are given below.

Reviewer #1:

Comment 1: in the method part, Line 77, it is better to provide the location where the E. pela is collected.

Response: We have added the location information “We used male adults of E. pela reared at the Institute of Highland Forest Science Kunming City (25°05′N, 102°72′E), Yunnan Province, for the mitochondrial genome sequencing” in this revision.

Comment 2: in Figure 1, the length of the mitochondrial genome is 17,766bp, not 177,66bp.

Response: Thanks for pointing it out. We have revised this mistake in Figure 1.

Comment 3: line 185, the 3 most divergent regions.

Response: We have replaced the word “different” with “divergent” in this revision.

Comment 4: Line 188, for the obvious insertions between rrnS-nad1, and between cox1-trnL1, and the deletion between...

Response: We have revised the specific word as you suggested.

Comment 5: why do you label atp8 by redbox in Figure 2?

Response: Compared with the other 3 Coccoidea species and D. melanogaster in Figure 2 (whose atp8 and atp6 were adjacent genes), E. pela’s atp8 was in the most unusual position, for which we believed it occurred an obvious rearrangement, so we labeled E. pela’s atp8 with red box in Figure 2.

Comment 6: in the discussion part, since the nuclear genome of E. pela has been published, is there any similar finding that A+T content also increased in the nuclear genome? 

Response: The assembled genome size of E. pela is 660 Mb with 66.2% AT content. The mitochondrial genome of E. pela (17,766bp) is just a very small part of the whole genome, and we thought it almost has no effect on AT content of the nuclear genome. Thank you for your thoughtful comments.

Reviewer 2 Report

The authors reported the complete mitochondrial genome of the Chinese white wax scale insect, Ericerus pela Chavannes, and reconstructed the phylogeny tree. The analyses results showed E. pela had significant gene rearrangements involving tRNAs compared with other Coccoidea species. And E. pela’s 9 tRNAs were identified to have obvious truncated structures. The findings from this study have great significance for determining E. pela’s phylogenetic position and would further help in the evolutionary studies of Coccidae. The manuscript is generally well written. However, there are some problems to be further improved as well. I would recommend accept with major revisions.

Line 14: “Chavannes” should be regular.

Line 77: Please add the collection site coordinates.

Line 92-105: The procedures are chaotic.

Please rephrase the two paragraphs following the steps below:

Ø  Library construction (The “library construction” content is lacked now)

Ø  Sequencing

Ø  Quality control

Ø  Genome assembly

Line 94, 102, 103: There is no need to abbreviate the “circular consensus sequence”.

Line 116, 120, 155, 236, ……: Please make the version presentation consistent.

Line 123-130: I think the Cicadoidea should be treated as ingroup. Species from Heteroptera may be selected as the outgroup.

Line l34: “MEG” should be “MEGA”

Line 135: MrBayes and one of the following two programs, RaxML or IQ tree, are recommended to construct the phylogenetic tree.

Line 143-145: The species scientific names should be italic.

Line 142-149: Although the PacBio platform can produce long reads, the experimental verification was recommended to do for the gene rearrangement and tRNAs truncation.

Line: 268-269: There is no need to put the “adenine”, “thymine”, “cytosine” and “guanine” here.

Line 279-283: In a way, I don’t think MITOS (Al Arab et al.) provided another successful precedent for predicting genes with high A+T contents, for MITOS (Al Arab et al.) has been used all the time. Please clarify.

Author Response

Response to Reviewers

Dear Reviewers:

We appreciate your comments regarding our manuscript. Based on your suggestions, we amended the relevant parts of the manuscript as much as possible. Point-by-point responses to your comments are given below.

Reviewer #2:

The authors reported the complete mitochondrial genome of the Chinese white wax scale insect, Ericerus pela Chavannes, and reconstructed the phylogeny tree. The analyses results showed E. pela had significant gene rearrangements involving tRNAs compared with other Coccoidea species. And E. pela’s 9 tRNAs were identified to have obvious truncated structures. The findings from this study have great significance for determining E. pela’s phylogenetic position and would further help in the evolutionary studies of Coccidae. The manuscript is generally well written. However, there are some problems to be further improved as well. I would recommend accept with major revisions.

Response: Thank you very much. We appreciate your sincere words and thoughtful suggestions. Based on your suggestion we have made corrections in the manuscript as much as possible.

Comment 1: Line 14: “Chavannes” should be regular.

Response: “Chavannes” is revised and shown in regular now.

Comment 2: Line 77: Please add the collection site coordinates.

Response: We have added the location information “We used male adults of E. pela reared at the Institute of Highland Forest Science Kunming City (25°05′N, 102°72′E), Yunnan Province, for the mitochondrial genome sequencing” in the revision.

Comment 3: Line 92-105: The procedures are chaotic.

Please rephrase the two paragraphs following the steps below:

Ø  Library construction (The “library construction” content is lacked now)

Ø  Sequencing

Ø  Quality control

Ø  Genome assembly

Response: We have added the “library construction” content and rephrased the procedures as shown in two paragraphs following the steps you mentioned above.

Comment 4: Line 94, 102, 103: There is no need to abbreviate the “circular consensus sequence”.

Response: We have revised “CCS” as “circular consensus sequence” in this revision.

Comment 5: Line 116, 120, 155, 236, ……: Please make the version presentation consistent.

Response: We have revised line 116 to be “Proksee (http://Proksee.ca/projects)”.  For line 120, we have checked that the UTL and version of RNA Folding Form, there was no mistake. For line 155 we have added the UTL of NCBI (https://www.ncbi.nlm.nih.gov/) in the revsion. For line 236, we have added the UTL of MITOS (http://mitos2.bioinf.uni-leipzig.de/index.py).

Comment 6: Line 123-130: I think the Cicadoidea should be treated as ingroup. Species from Heteroptera may be selected as the outgroup.

Response: We have added the Heteroptera species Lethocerus deyrollei (GenBank: KU237288.1) as the outgroup in the phylogenetic analysis. The Cicadoidea is treated as ingroup now.

Comment 7: Line l34: “MEG” should be “MEGA”

Response: We have replaced the original MEGA tree with the new tree made by IQTREE, and revised the mistake in this revision.

Comment 8: Line 135: MrBayes and one of the following two programs, RaxML or IQ tree, are recommended to construct the phylogenetic tree.

Response: We have redrawn the phylogenetic tree with the program IQTREE and the best suitable model was calculated by the program itself. With the lowest score of the Bayesian Information Criterion (BIC), the model “mtZOA+F+I+G4” was selected in the analysis. Also, we constructe another tree by MrBayes (http://www.phylogeny.fr/one_task.cgi?task_type=mrbayes) with the same species used in IQTREE. We have replaced the original tree with the IQTREE in this revision.

IQ-TREE

MrBayes-tree

Comment 9: Line 143-145: The species scientific names should be italic.

Response: We have revised species names as shown in italic.

Comment 10: Line 142-149: Although the PacBio platform can produce long reads, the experimental verification was recommended to do for the gene rearrangement and tRNAs truncation.

Response: As you mentioned, PacBio sequencing could produce long reads, and the HiFi model of PacBio sequence has accuracy >99%. Experimental detection could no doubt verify the existing results. However, considering the low error rate of HiFi model, we thought the annotation result of E. pela was credible as well.

Comment 11: Line: 268-269: There is no need to put the “adenine”, “thymine”, “cytosine” and “guanine” here.

Response: We have deleted the words you mentioned in this revision.

Comment 12: Line 279-283: In a way, I don’t think MITOS (Al Arab et al.) provided another successful precedent for predicting genes with high A+T contents, for MITOS (Al Arab et al.) has been used all the time. Please clarify.

Response: As you can see on MITOS webserver homepage, the method of AL Arab et al. is not a default option, it is an advanced parameter. According to previous research, PCGs of E. pela could not be completely detected without applying this method. And after checking this option on MITOS, all the 13 PCGs were successfully predicted. Also, with referring to the paper [1] which mentioned “the method of AL Arab et al. [2] can detect unannotated frameshifts and misannotated genes of metazoan mitogenomes based on enhanced phylogeny-aware hidden Markov models (HMMs). All PCGs of the S. coffeae mitogenome were successfully annotated using the method of AL Arab et al.”, so we thought our study provided another successful precedent for predicting genes with high A+T contents.

(mitogenome of S. coffeae also contains a high AT content)

References:

[1] Lu C, Huang X, Deng J. The challenge of Coccidae (Hemiptera: Coccoidea) mitochondrial genomes: The case of Saissetia coffeae with novel truncated tRNAs and gene rearrangements. Int J Biol Macromol. 2020;158:854-864.

[2] Al Arab M, Höner Zu Siederdissen C, Tout K, Sahyoun AH, Stadler PF, Bernt M. Accurate annotation of protein-coding genes in mitochondrial genomes. Mol Phylogenet Evol. 2017;106:209-216.

Reviewer 3 Report

This paper is a very simple presentation of a novel hemipteran mitogenome and some interesting phylogenetic analysis. Overall, it is well done and adequately presented. The authors did a good job of discussing specific issues in the Discussion that arose during my reading of the Methods and Results.

There are some grammar issues throughout that will need to be correct prior to publication, as well as a few other minor issues that will need to be addressed.

1. Table 1 - Format for some cells is not wide enough for the full words. i.e. Aleyrodoidea

2. Section 2.7 - Species names are not italicized.

3. Sections 2.7, 3.2, & 4.3 - As the authors note, D. melanogaster may not be the best comparison for the E. pela gene arrangement and synteny analyses (though it should be included simply due to the robustness of the annotation). I would like to see inclusion of an additional model Hemipteran species, if available, (i.e. pea aphid, A. pisum) that might be a good intermediary between the D. mel and the focal species.

Figure 5 - The branch lengths and node scores can be difficult to read and sometimes overlap, making the figure difficult to read. Please revise.

Section 4 - There is very little discussion of the phylogenetic analysis. I would like to see an expansion of this section that addresses the higher level phylogenetics of this group.

Author Response

Response to Reviewers

Dear Reviewers:

We appreciate your comments regarding our manuscript. Based on your suggestions, we amended the relevant parts of the manuscript as much as possible. Point-by-point responses to your comments are given below.

Reviewer #3:

This paper is a very simple presentation of a novel hemipteran mitogenome and some interesting phylogenetic analysis. Overall, it is well done and adequately presented. The authors did a good job of discussing specific issues in the Discussion that arose during my reading of the Methods and Results.

There are some grammar issues throughout that will need to be correct prior to publication, as well as a few other minor issues that will need to be addressed.

Response: Thank you very much. We appreciate your sincere words and thoughtful suggestions. Based on your suggestion we have made corrections in the manuscript as much as possible.

Comment 1: Table 1 - Format for some cells is not wide enough for the full words. i.e. Aleyrodoidea.

Response: We have widen the column of Table 1 in this revision.

Comment 2: Section 2.7 - Species names are not italicized.

Response: We have revised species names as they are shown in italic now.

Comment 3: Sections 2.7, 3.2, & 4.3 - As the authors note, D. melanogaster may not be the best comparison for the E. pela gene arrangement and synteny analyses (though it should be included simply due to the robustness of the annotation). I would like to see inclusion of an additional model Hemipteran species, if available, (i.e. pea aphid, A. pisum) that might be a good intermediary between the D. mel and the focal species.

Response: To maintain the consistency of the species that to be analyzed, we have added the species Aphis glycines (Hemiptera: Aphididae) in rearrangement analysis. For systeny analysis, we conducted the analysis only in Coccoidea (not including D. melanogaster) to find out the mitochondrial genetic varieties in Coccoidea, so we thought it was better not to add A. glycines in this part.

Comment 4: Figure 5 - The branch lengths and node scores can be difficult to read and sometimes overlap, making the figure difficult to read. Please revise.

Response: We have reconstructed the phylogenetic tree (figure 5), and as you suggested, the branch lengths are shown in larger word size than node scores now.

Comment 5: Section 4 - There is very little discussion of the phylogenetic analysis. I would like to see an expansion of this section that addresses the higher level phylogenetics of this group.

Response: We have expanded discussion regarding phylogenetic analysis and higher level of classification in 4.4.

Round 2

Reviewer 2 Report

The authors have revised the manuscript according to the comments, and an acceptance is suggested.

Author Response

Response to Reviewers

Reviewer #2:

The authors have revised the manuscript according to the comments, and an acceptance is suggested.

Response: We appreciate your sincere words and thoughtful suggestions for our work. Thank you so much.